# Deep Learning-Based Dynamic Computation Task Offloading for Mobile Edge Computing Networks

**DOI:** 10.3390/s22114088

**Published:** 2022-05-27

**Authors:** Shicheng Yang, Gongwei Lee, Liang Huang

**Affiliations:** 1The College of Information Engineering, Zhejiang University of Technology, Hangzhou 310023, China; scyang@zjut.edu.cn; 2The College of Computer Science and Technology, Zhejiang University of Technology, Hangzhou 310023, China; gwlee@zjut.edu.cn

**Keywords:** mobile-edge computing, deep learning, computation offloading

## Abstract

This paper investigates the computation offloading problem in mobile edge computing (MEC) networks with dynamic weighted tasks. We aim to minimize the system utility of the MEC network by jointly optimizing the offloading decision and bandwidth allocation problems. The optimization of joint offloading decisions and bandwidth allocation is formulated as a mixed-integer programming (MIP) problem. In general, the problem can be efficiently generated by deep learning-based algorithms for offloading decisions and then solved by using traditional optimization methods. However, these methods are weakly adaptive to new environments and require a large number of training samples to retrain the deep learning model once the environment changes. To overcome this weakness, in this paper, we propose a deep supervised learning-based computational offloading (DSLO) algorithm for dynamic computational tasks in MEC networks. We further introduce batch normalization to speed up the model convergence process and improve the robustness of the model. Numerical results show that DSLO only requires a few training samples and can quickly adapt to new MEC scenarios. Specifically, it can achieve 99% normalized system utility by using only four training samples per MEC scenario. Therefore, DSLO enables the fast deployment of computation offloading algorithms in future MEC networks.

## 1. Introduction

The rapid development of wireless communication technologies has driven the emergence of more and more latency-sensitive and resource-intensive applications and services, such as augmented reality, voice recognition, image recognition, and mobile health care. These mobile applications have dramatically increased the demand for computing and storage resources for wireless devices, which are typically provided by cloud servers. However, because cloud servers are usually deployed over long distances, high transmission latency results when tasks are offloaded to the cloud. Therefore, mobile edge computing (MEC) [1] is considered an effective solution to this problem. The fundamental principle of MEC is that by deploying hosts with cloud computing capabilities at the edge of the wireless network. wireless devices can offload tasks to edge servers, solve the problem of insufficient computational resources, reduce the delay of computing tasks [2,3], and save energy consumption [4,5].

As one of the key technologies of MEC, computation offloading solves the problem of resource limitation and low processing efficiency by offloading computation-intensive tasks to MEC hosts at the network edge. On the one hand, the task offloading process is affected by different factors, e.g., computational resources, wireless communication quality, etc. On the other hand, if computational tasks are aggressively offloaded to the edge servers, the bandwidth in the network system will be occupied. The uplink wireless channel will be severely congested, which significantly increases the transmission delay of computational tasks. Therefore, how to make the best decision is the critical issue of edge offloading. In general, computation offloading and resource allocation are jointly formulated as nonlinear programming problems with mixed-integer programming. Most of the existing solutions are based on heuristic or approximation algorithms [6,7]. However, these solutions rely on expert knowledge and exact mathematical models, which often require a fresh start to adjust the mathematical models once the MEC environment changes, resulting in inefficient offloading decisions. The computational complexity of such methods is relatively high. Therefore, it remains a challenge to design an MEC network with a low-complexity algorithm that can be applied to time-varying environments.

Recently, deep learning has been widely used to optimize offloading decisions [8]. Since the nature of a neural network is a black-box model that does not require precise expert knowledge and accurate mathematical models, deep learning provides a good idea for the above challenges. Deep reinforcement learning learns to solve complex problems, such as video games, chess activities, and intelligent robot control, through continuous trial and error. Currently, computational offloading based on deep Q-learning (DQL) [9] is most widely studied, which discretizes the state/action space and optimizes computational offloading decisions and system resource optimization through online learning. However, the discretization of continuous variables limits the performance of DQL and is not suitable for dealing with high-dimensional action spaces [10]. In addition, most of these methods are based on static interactive MEC environments with sufficient training samples to train neural networks. Once the MEC scene changes, it is difficult to collect enough training samples when converging to a new scene. However, considering the dynamic nature of the offloading task, how to converge to a new scene quickly with small samples is still a problem worth investigating.

There are some studies on dynamic MEC networks in the literature. Huang et al. [11] constructed dynamic MEC scenarios by time-varying wireless channel gains and weight factors of computational tasks and proposed a meta-learning-based computation offloading (MELO) algorithm to obtain a general model that can be quickly adapted to new task scenarios by a small number of training samples in MEC scenarios. Wang et al. [12] constructed dynamic task scenarios by different topologies and proposed a meta-reinforcement learning-based approach to improve the training efficiency and reduce the dependence on samples. Unfortunately, this research ignores the energy consumption of the system to achieve low latency. Moreover, the training process of the computational offloading algorithm based on meta-learning is tedious. A new model needs to be created additionally for computing the gradient at each training.

In this paper, we consider a multi-scenario MEC network with multiple WDs and a single edge server, where each MEC scenario is characterized by a specific task weighting factor. Each WD in the network makes real-time decisions on whether to offload its computing tasks to the edge server or execute it locally. By jointly optimizing WDs’ computing offload and bandwidth allocation, we propose a Deep-Supervised-Learning-based computation Offloading (DSLO) algorithm to learn to minimize the network utility. The main contributions of this paper are as follows:We model the system utility of the MEC network with prioritized computing tasks such as the weighted sum of energy consumption and delay cost. To minimize the system utility, we decompose the joint optimization problem into the offloading decision subproblem and the transmission bandwidth allocation subproblem, which are further solved via deep learning and optimization methods, respectively.We propose the DSLO framework to learn from a few training samples to optimize the offloading decision actions. We introduce the batch normalize (BN) layer in CNN/DNN network structure to accelerate the convergence process. It can efficiently learn the mapping from the workload and weight factors to computational offloading.Simulation results show that DSLO-CNN can generate near-optimal offloading decisions and outperforms DSLO-DNN under MEC scenarios training datasets of different sizes. Significantly, the normalized system utility of the DSLO-CNN algorithm achieves a median value of 96% when only 10% of MEC scenarios are included in the training dataset with two training samples per MEC scenario. In new MEC scenarios, DSLO-CNN converges faster than MELO.

The remainder of this paper is organized as follows. The related work is introduced in Section 2. In Section 3, we present the system model and problem formulation. We propose the DSLO algorithm in Section 4. Numerical results are presented in Section 5, and a conclusion is provided in Section 6. Before leaving this section, those important notations and abbreviations used throughout this paper are respectively summarized in Abbreviations.

## 2. Related Work

So far, considerable research efforts have been devoted to the offloading scheme design for MEC networks. The computing offloading optimization problem can be modeled by three elements [13], i.e., state parameters, decision actions, and system utility. The current state of research on computational offloading can be divided into static MEC network computational offloading and dynamic MEC network computational offloading, depending on the number of state parameters.

Under static MEC network computational offloading [14,15,16], the offloading decision is optimized for a unique category of state parameters, such as wireless channel gain or task computation amount. You et al. [17] considered a single-user system with wireless channel gain as the state parameter and optimized the performance of local computing and offloading computing under the constraints of energy harvesting and processing latency. Tran et al. [18] proposed a low-complexity heuristic offloading framework with the wireless channel gain as the state parameter. Bi et al. [19] considered a cache-assisted MEC system, where the server can selectively cache the previously generated programs for future reuse. Resorting to the deep neural network (DNN), Huang et al. [20] proposed a deep reinforcement learning-based computational offloading framework (DROO), which maximizes the system computation rate by jointly optimizing computation offloading and resource allocation according to the time-varying channel gains. Considering the static interaction environment of MEC, we believe that there are sufficient training samples under the static MEC network. Most of the offloading algorithms based on deep learning are based on DNN.

Considering the dynamic properties of wireless applications, MEC networks make computational offloading decisions by evaluating multiple classes of state parameters jointly. Min et al. [21] considered a dynamic MEC network consisting of a time-varying radio link transmission rate and investigated the computational offloading of IoT devices with energy harvesting in a dynamic MEC network. They proposed an offloading scheme based on deep reinforcement learning, which uses a convolutional neural network (CNN) to compress the state space to speed up the convergence process of the algorithm. Huang et al. [11] considered a dynamic computing task scenario consisting of different weight priority coefficients and proposed a meta-learning-based computing offloading (MELO) framework. MELO can efficiently adapt to a new MEC scenario and minimize the total system delay. Qu et al. [22] considered a dynamic computing task scenario composed of different computing power and bandwidth and proposed a computing offloading framework based on meta-reinforcement learning. The algorithm can quickly adapt to complex and dynamic environments and can be used to improve the robustness of task offloading decisions in IoT environments. Wang et al. [12] constructed dynamic computing task scenarios based on different network topologies, modeled the computational tasks as directed acyclic graphs (DAGs), and designed a sequence-to-sequence (seq2seq) neural network model to generate offloading strategies that can quickly adapt to new environments. Chen et al. [23] considered an MEC system composed of a random task arrival model, where the state parameters include the size of the input data, the maximum tolerable delay, the number of CPU cycles, and the time slot of the task arrival. They designed a temporal feature extraction network composed of one-dimensional convolutional (Conv1D) residual blocks and a long short-term memory (LSTM) network to solve the joint optimization problem of computational offloading and resource allocation. For dynamic MEC network computational offloading, the state parameters are too complicated to optimize the decision actions via classical optimizations, where computation offloading and resource allocation are jointly formulated as a mixed-integer nonlinear programming problem [24]. Considering the dynamics and complexity of the actual MEC environment, we assume that the modeled dynamic MEC scenario contains only a small number of training samples.

Recent works [12,13,20,23,25] resort to the deep learning methods, which utilize deep neural networks [26] to learn from data samples and generate the optimal mapping from state space to action space. However, they require a large number of data samples to train the neural network and cannot converge to the optimal offloading performance when lacking training samples. Achieving efficient deep learning-based computing offloading under a few training samples is challenging in dynamic MEC networks. To the best of our knowledge, there are relatively few studies investigating binary offloading designs based on dynamic MEC networks. Unlike the literature [11], the objective of the study in this paper also includes energy consumption and weight factors as input data together with the neural network and uses CNN networks for offloading prediction. Therefore, our work focuses on designing binary offloading strategies based on time-varying workloads and different computational tasks, aiming to minimize the weighted sum of energy consumption and time delay.

## 3. System Model

We consider an MEC network composed of one edge server, one wireless access point (AP), and *N* WDs, denoted as N={1,2,⋯,N}, as shown in Figure 1. The AP and the edge server are connected by optical fiber, whose transmission delay can be ignored. Each WD has one task that needs to be processed locally or be offloaded to the edge server through the AP. Without loss of generality, each WD needs to execute one prioritized computing task with a specific task weight. Denote $w_{n}\in W$ as the task weight of $WD_{n}$, whose value changes in the countable set W depending on the specific task category. An MEC task scenario is considered different from another one if at least one task’s weight is different. Each WD can decide whether to offload its tasks to edge servers or compute locally by the device itself. Denote $a=\{a_{n}\in \left\{0,1\right\}|n\in N\}$ as the offloading decision. Specifically, $a_{n}=1$ indicates that the $WD_{n}$ offloads its task to the edge server and $a_{n}=0$ means its task is processed locally.

### 3.1. Energy Consumption

When the offloading decision is present, we study the resource allocation optimization problem of the MEC network and model dynamic computation tasks via task weights. We set a tuple $\left(d_{n},\gamma_{n}\right)$ to represent $WD_{n}$’s task, for n∈N. Specifically, $d_{n}$ is the workload of $WD_{n}$, and $\gamma_{n}$ is the required number of CPU cycles to complete the task. When $WD_{n}$ offloads its total task to the edge server, the energy consumption includes data transmission and task computation, which can be expressed as
(1)$$\begin{array}{c} E_{n}^{c}=E_{n}^{t}+\alpha d_{n}, \end{array}$$
where $E_{n}^{t}$ is the transmission energy consumed by $WD_{n}$ for uploading its workload to the edge server, which is a linear function of workload $d_{n}$, $\alpha d_{n}$ is the task computation energy consumption, and α is the factor of the computation energy consumption at the edge server. In the above formula, superscripts *c* and *t* stand for “edge computing” and “transmission”, respectively. When α=0, we only consider the transmission energy consumption of WDs. Here, we ignore the energy consumption and delay of the edge server when transmitting the computation results back to $WD_{n}$. Similar to the literature [27], we assume that the computed result size is much smaller than the input data size.

When $WD_{n}$ executes its task locally, we define $e_{n}^{l}$ as the local energy consumption per data bit of $WD_{n}$, where the superscript indicates the offloading method. Here, the superscript is *l* denotes “local computing”. So, $WD_{n}$’s energy consumption for executing its total task locally can be given by:(2)$$\begin{array}{c} E_{n}^{l}=d_{n}e_{n}^{l}. \end{array}$$

By evaluating the energy consumption of both computation offloading and local execution under the offloading decision $a_{n}$, we get the total energy consumption of $WD_{n}$, as
(3)$$\begin{array}{c} E_{n}=E_{n}^{c}a_{n}+E_{n}^{l}\left(1-a_{n}\right). \end{array}$$

### 3.2. Time Delay

In addition to the total energy consumption, another major factor affecting the system’s overall efficiency is the time delay. It includes the transmission delay when WDs offload tasks to MEC servers and the processing delay of MEC servers, and WDs’ local execution latency.

Considering the delay in computation offloading, when $WD_{n}$’s task is offloaded to the edge server, we use $c_{n}$ to denote the allocated bandwidth to $WD_{n}$ for task transmission. Then, the time latency $T_{n}^{c}$ can be expressed as
(4)$$\begin{array}{c} T_{n}^{c}=\frac{d_{n}}{c_{n}}+\frac{\gamma_{n}}{f_{n}}. \end{array}$$

The transmission delay of $WD_{n}$ is defined as $\frac{d_{n}}{C_{n}}$, and the computational delay of $WD_{n}$ is defined as $\frac{\gamma_{n}}{f_{n}}$, where $f_{n}$ is defined as the edge processing rate.

Meanwhile, the local execution delay for $WD_{n}$ to execute its task is given by
(5)$$\begin{array}{c} T_{n}^{l}=d_{n}t_{n}^{l}, \end{array}$$
where $t_{n}^{l}$ is the local execution delay per data bit of $WD_{n}$.

Therefore, the total time delay is a combination of delay under both computation offloading and local execution, which can be given as
(6)$$\begin{array}{c} T_{n}=T_{n}^{c}a_{n}+T_{n}^{l}\left(1-a_{n}\right). \end{array}$$

### 3.3. Problem Formulation

We formulate the joint computation offloading and bandwidth allocation for the MEC network as an mixed integer optimization problem. We introduce the system utility $Q\left(d,a,w,c\right)$ to represent the sum cost of the MEC network as:$$Q\left(d,a,w,c\right)=\sum_{n=1}^{N}E_{n}w_{n}+\beta \max\left\{T_{n}w_{n}|\forall n\in N\right\},$$
where $d=\left\{d_{n}|n\in N\right\}$, $a=\left\{a_{n}|n\in N\right\}$, $c=\left\{c_{n}|n\in N\right\}$, and β denotes the weight of energy consumption and task completion. Considering dynamic MEC scenarios with different computation task d and weights w, we aim to optimize offloading decisions a and bandwidth allocation c to minimize the system utility Q(d,a,w,c). The optimization problem can be defined as (P1): (7)$$\begin{array}{cc} \left(P1\right):\;Q^{*}\left(d,w\right)=\; & \underset{a,c}{\min\;}Q\left(d,a,w,c\right) \end{array}$$(8)$$\begin{array}{cc} s.t.\;\;\;\;\;\;\;\; & c_{n}\geq 0,\;\forall n\in N, \end{array}$$(9)$$\begin{array}{cc} \; & \sum_{n=1}^{N}c_{n}\leq C, \end{array}$$(10)$$\begin{array}{cc} \; & w_{n}\in W, \end{array}$$(11)$$\begin{array}{cc} \; & a_{n}\in \left\{0,1\right\}. \end{array}$$

Here, the constraints (8) and (9) mean that the bandwidth allocated to all WDs $c_{n}$ is non-negative, and *C* limits the total bandwidth. Since the constraints on the offloading decision and bandwidth allocations are decoupled, we decompose problem (P1) into an offloading decision subproblem and a bandwidth allocation subproblem (P2), as illustrated in Figure 2.

Considering the continuously changing data size $d_{n}$ and the occasionally changing task weight $w_{n}$, the offloading decision subproblem aims to find an offloading strategy π to effectively produce the optimal offloading decision $a^{*}$, as:(12)$$\begin{array}{c} \pi :d_{n}\mapsto a_{n}^{*}. \end{array}$$

The subproblem (P2) aims to optimize the bandwidth allocation c and is expressed as
$$\begin{array}{cc} \left(P2\right):\;Q^{*}\left(d,w\right)=\; & \min_{c}\;Q\left(d,a,w,c\right)\\ s.t.\;\;\;\;\;\;\;\; & (9)\;\mathrm{and}\;(8). \end{array}$$

Given $a^{*}$, the subproblem (P2) is convex and can be efficiently solved by the standardized CVX tool package.

We assume that changes in workload d are faster than changes in task weights w due to the heterogeneity of mobile devices. The MEC scenario occasionally changes when the task weight changes. Some existing deep learning-based computational offloading methods are based on static network scenarios and require a large amount of training data samples. Once the MEC scenario changes, the deep learning model is difficult to adapt to a new scenario. In the next section, we propose an algorithm based on deep learning that uses few training samples and can quickly adapt to new MEC scenarios.

## 4. Deep Supervised Learning-Based Offloading Algorithm

In this section, we propose a DSLO algorithm to achieve the optimal binary offloading decision. The algorithm takes advantage of the translation invariance of the convolutional neural network to capture the local features of input data and accelerate the convergence of the offloading process. For dynamic MEC task scenarios, DSLO can quickly adapt to the new workload d and task weights w. The overall pipeline of the DSLO algorithm is illustrated in Figure 3.

We consider a dynamic MEC task scenario, where the task weight w of the scenario is variable and the task workloads d can be changed independently. The dataset contains I={1,2,…,L} different MEC task scenarios $\left\{\Psi_{i}|i\in I\right\}$, where each MEC task scenario $\Psi_{i}$ contains K={1,2,⋯,K} data samples, denoted as $\Psi_{i}=\left\{{\left(d_{k},w_{k},a_{k}\right)}^{i}|k\in K\right\}$. Here, each data sample $\left(d_{k},w_{k},a_{k}\right)$ is a combination of workload, weight factor, and optimal offloading decision. The dataset is further randomly split into $I_{train}$ for the training phase and $I_{test}$ for the testing phase. Correspondingly, we denote $K_{train}$ and $K_{test}$ as the index set of data samples for training and testing under each MEC scenario.

### 4.1. Neural Network Architecture

We implement DSLO based on two classical neural network architectures, CNN and DNN. As shown in Figure 4, the DSLO-CNN algorithm model utilizes three convolutional layers and three fully connected layers. Each layer is followed by the rectified linear (ReLU) activation function except for the output layer. The Sigmoid activation function is used to drop the output value near 0 or 1. The DSLO-CNN algorithm captures the associated information of $d_{i}$ and $w_{i}$ via block by block scanning and focuses on local contents.

The DSLO-DNN algorithm structure is simpler and only uses fully connected layers, as shown in Figure 5. It weights and processes all data through the full connection. Although the influence of all data on a single node is considered, the association information between workloads d and task weights w is not highlighted.

Table 1 is a parameter description of both DLSO algorithm structures. Each convolution kernel has a size of 2. BN denotes batch normalization whose size depends on the size of the corresponding layer. We set ReLU activation functions at each layer except the output layer in order to increase the generalization performance of the model and alleviate the problem of overfitting in a single scenario.

We use BN to address issues related to internal covariance shifts in feature diagrams, so as to prevent model overfitting by smoothing the flow of gradients and improving network generalization. Its mathematical expression is as follows:(13)$$\begin{array}{c} \left\{\begin{array}{c} {\hat{y}}_{i}=\gamma \frac{x_{i}-u}{\sqrt{\sigma^{2}+\varepsilon }}+\beta ,\\ u=\frac{1}{S}\sum_{i=1}^{S}x_{i},\\ \sigma^{2}=\frac{1}{S}\sum_{i=1}^{S}{\left(x_{i}-u\right)}^{2}, \end{array}\right. \end{array}$$
where *S* denotes the batch size, $x_{i\epsilon S}$ denotes the input data, ${\hat{y}}_{i}$ denotes the data after BN, *u* and σ are the means. Regarding the variance of input data, γ and β are the scale factor and the offset, respectively. To avoid a denominator of 0, ε is usually set to $1.0\times 10^{-5}$ to increase numerical stability [28].

### 4.2. Train DSLO

To train DSLO, we randomly sample a batch of MEC scenarios $\left\{\Psi_{i}|i\in I_{train}\right\}$, where each scenario $\Psi_{i}$ contains $K_{train}$ data samples where $K_{train}\subset K$. We merge these training samples from different MEC scenarios together, denoted as $M=\left\{{\left(d_{k},w_{k},a_{k}\right)}^{i}|k\subset K_{train},i\subset I_{train}\right\}$. During each round of training, we randomly sample a set of training data samples $M_{b}$ from M, denoted as $D=\left\{\left(d_{m},w_{m},a_{m}\right)|m\in M_{b}\right\}$. We use randomly sampling with replacement to prevent the model from being overly dependent on a particular batch of data during training and thus falling into a local optimum [29]. Then, we train the neural network model by minimizing its mean-squared error loss [30] as
(14)$$L\left(f_{\theta }\right)=\sum_{m\in M_{b}}{∥f_{\theta }\left(d_{m},w_{m}\right)-a_{m}∥}_{2}^{2},$$
where $f_{\theta }$ is a parameterized function of the model and represents the mapping relationship between computation task and offloading decision. Then, the model’s parameters θ are updated by gradient descent [31], i.e.,
(15)$$\theta^{\prime }=\theta -\eta \nabla_{\theta }L\left(f_{\theta }\right),$$
where $\nabla_{\theta }$ is calculated with respect to the gradient of θ, and η is a step length hyper-parameter. After *G* rounds of training iteration, the network model converges and is used as the offload decision prediction model for a dynamic MEC network composed of different workloads and weight factors.

### 4.3. Test DSLO

During the test phase, for each MEC scenario in the test dataset $\left\{\Psi_{i^{\prime }}|i^{\prime }\in I_{test}\right\}$, we randomly sample a batch of data samples $K_{test}\subset K$ from $\Psi_{i^{\prime }}$. For each test data sample, we input the workload $d_{k^{\prime }}^{i^{\prime }}$ and weighting factors $w_{k^{\prime }}^{i^{\prime }}$ into the DSLO and obtain the predicted offloading decision $\hat{a}$, as
(16)$$\hat{a}=f_{\theta^{\prime }}\left(d_{k^{\prime }}^{i^{\prime }},w_{k^{\prime }}^{i^{\prime }}\right),k^{\prime }\subset K_{test},i^{\prime }\in I_{test}$$

Finally, based on $\hat{a}$, we evaluate the network utility *Q* by solving the subproblem (P2).

Before leaving this section, we provide the pseudo-code of the DSLO algorithm in Algorithm 1.
**Algorithm 1:** Pseudo-code of the DSLO Algorithm.**Input   :** Dataset I of different MEC scenarios and step-size hyper-parameter η**Output:** The trained neural network modelRandomly initialize θRandomly split I into Itrain and ItestFor each scenario, randomly split its data samples K into Ktrain and KtestMerge all training samples into a whole training set M// Training procedure
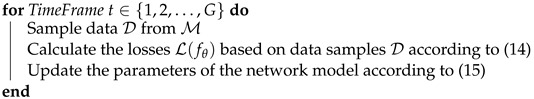
// Testing procedureGiven a new sample set from Ktest, generate its offloading decision a^Evaluate the network utility *Q* by solving the subproblem (P2)

## 5. Performance Evaluation

### 5.1. Parameter Settings

In this section, we study the numerical performance of the proposed DSLO algorithm under different MEC scenarios. In the simulation, we follow the settings in [27] and set the local computation time of the mobile device as $t_{n}^{l}=4.75\times 10^{-7}$ s/bit and the processing energy consumption as $e_{n}^{l}=3.25\times 10^{-7}$ J/bit. We assume that the input data size of all tasks $d_{n}$ is randomly distributed between 10 and 30 MB. In addition, we consider x264 video encoding application as the computing task of WDs where the number of computational cycles is relevant to the input size as $\gamma_{n}=1900$ cycles/byte × $d_{n}$ [32]. The transmission energy consumption of mobile devices is, respectively, $e_{n}^{t}=1.42\times 10^{-7}$ J/bit. The CPU rate of the edge server is $10\times 10^{9}$ cycles/s. We further set $\alpha =3.5\times 10^{-7}$ J/bit and the uplink bandwidth limit as 100 Mbps. In Table 2, we summarize the simulation parameters for easy reference.

To train and evaluate the DSLO algorithm, we pre-generate a dataset composed of various MEC task scenarios. For an MEC task scenario, given the workload $d_{t}$ and the task weight $w_{t}$ at time *t*, we compute its best offloading action by enumerating all $2^{N}$ binary combinations, calculate all those corresponding optimization utilities $Q_{t}$ by solving subproblem (P2), and choose the offloading decision $a_{t}^{*}$ with the minimum utility $Q_{t}*$ as the optimal decision. Then, we add the state parameter $\left(d_{t},w_{t}\right)$ and the offloading decision along with its optimal $\left(a_{t}^{*},Q_{t}*\right)$ into the dataset. Considering extending to generality, we set the task weights W={1,1.5}. We evaluate an MEC network with N=10 WDs and include all 1024 MEC task scenarios with different weights in the dataset, which are further used to train and test DSLO.

Regarding network utility performance, we compare our DSLO algorithm with three representative benchmarks:1*Random offloading decision*: All *N* WDs randomly generate 0–1 offloading decisions.2*Linear Relaxation (LR) algorithm* [33]: The binary offloading decision variable conditioned on (11) is relaxed to a real number between 0 and 1, as ${\hat{a}}_{n}\in \left[0,1\right]$. Then, the optimization problem (P1) with this relaxed constraint is convex with respect to $\left\{{\hat{a}}_{n}\right\}$ and can be solved using the convex optimization toolbox. Once ${\hat{a}}_{n}$ is obtained, the binary offloading decision $a_{n}$ is determined as follows
(17)$$\begin{array}{c} a_{n}=\left\{\begin{array}{cc} 1, & \;\mathrm{when}\;{\hat{a}}_{n}\geq 0.5\\ 0, & \;\mathrm{otherwise}\; \end{array}\right. \end{array}$$3*Greedy strategy*: For the greedy scheme, we enumerate all offloading decision combinations and then adopt the best one.

The following numerical results are an average of 200 realizations running on a laptop with Intel(R) Core(TM) i7-4710MQ CPU @ 2.50 GHz.

### 5.2. DSLO with Plenty Training Samples

We first evaluate the extreme performance of DSLO with a large amount of data samples. In Figure 6, we study the convergence performance of the DSLO algorithm for a specific MEC scenario, whose task weights are $w^{\prime }$= {1, 1.5, 1, 1.5, 1, 1.5, 1, 1.5, 1, 1.5}. In this scenario, we set $\left|M\right|=\left|K_{train}\right|=5000$, $\left|M_{b}\right|=128$, and $\left|K_{test}\right|=1000$. There are enough training samples to train DSLO until convergence.

For the sake of a better illustration, we define a normalized system utility:(18)$$\hat{Q}=\frac{Q^{*}\left(h,w,a^{*}\right)}{Q\left(h,w,a\right)}$$
which is a ratio between the enumerated optimal offloading action and the one generated by the evaluated algorithm. Both DSLO-CNN and DSLO-DNN converge to a normalized system utility $\hat{Q}\geq 99\%$. With regard to the convergence speed, DSLO-CNN only takes 1000 time frames, while DSLO-DNN takes 5000 time frames. Therefore, the DSLO-CNN algorithm can quickly converge and achieve approximately optimal convergence $\hat{Q}$.

In Figure 7, we compare the system utility performance of different offloading algorithms under the varying numbers of WDs. The MEC task scenarios are different due to the different number of WDs. For fairness, we have considered two different MEC scenarios: $\Psi_{1}$ and $\Psi_{2}$, whose weight factors are evenly distributed. For each WD, we give weight factors data in Table 3. Compare $\Psi_{1}$ and $\Psi_{2}$ with the data at the same position. For example, when *N* = 15 and CNN is adopted in the algorithm, the minimize system utility Q obtained, respectively, is 2230 and 2183, which is a small gap. To evaluate the extreme performance, both DSLO algorithms have been trained with 5000 independent workloads until convergence. Each value is an average of over 1000 independent test data samples in Figure 7. For different MEC scenarios, both DSLO-DNN and DSLO-CNN achieve approximately optimal performance as the Greedy algorithm and are significantly better than LR and Random algorithms. Under the condition of the same algorithm, only changing the task weight (priority relationship) between WDs will not produce a big difference in the results.

In Table 4, we compare the CPU execution latency of DSLO algorithms based on different WD numbers. The CPU execution latency of DSLO is significantly less than the widely used heuristic LR algorithm. When N=5, the CPU execution latency for LR to complete the given task is $4.14\times 10^{-2}$ s. The latency increases linearly with the number of WDs to $3.14\times 10^{-1}$ s for 15 WDs. In comparison, the test latency of DSLO is always below $10^{-3}$ s. While DSLO-CNN achieves better convergence performance due to its complex neural networks, it doubles the training latency of DSLO-DNN. Interestingly, the test latency of DSLO-CNN is almost the same as the one of DSLO-DNN, which generates an offloading within 1 millisecond.

In Figure 8, we evaluate the effect of BN in both DSLO algorithms. BN greatly improves the convergence performance of DSLO in metrics of both convergence speed and the optimal system utility. As shown in Figure 8a, with the BN structure, the training time of DSLO-CNN reduces from 4000 time frames to 1000 time frames. As shown in Figure 8b, models that do not contain BN layers have a high tendency to fall into local optima in the training phase.

### 5.3. DSLO with Few Training Samples

In Figure 9, we study the convergence performance of the DSLO algorithm for dynamic MEC scenarios with few training samples in each scenario. Specifically, we extract 70% of the entire scene from 1024 MEC task scenarios as the training set, i.e., $\left|I_{train}\right|=714$ and set $\left|M_{b}\right|=128$. For each MEC scenario, we only use $\left|K_{train}\right|=2$ and $\left|K_{test}\right|=100$. As shown in Figure 9, DSLO-DNN slowly converges to a normalized system utility $\hat{Q}=0.98$ until the 9000 time frames. In comparison, it converges to $\hat{Q}\geq 0.99$ within 4000 time frames when there are plenty of training samples, as shown in Figure 6. On the other hand, the DSLO-CNN algorithm remains in a high convergence speed, resulting in a normalized system utility $\hat{Q}\geq 99\%$ within 2000 time frames.

In Figure 10, we evaluate the convergence performance of DSLO with different scales of training scenarios. As shown in Figure 10a, when $\left|I_{train}\right|=10\%\left|I\right|$, we randomly choose 103 MEC scenarios out of 1024 scenarios to build the training dataset. Other system parameters are the same as the ones in Figure 9. It follows that there are only 206 training samples when $\left|I_{train}\right|=10\%\left|I\right|$. With such a few training samples, DSLO-CNN achieves a normalized system utility $\hat{Q}=98\%$ after convergence. As shown in Figure 10b, when $\left|I_{train}\right|=30\%\left|I\right|$, the DSLO-CNN algorithm converges within t=1000 with $\hat{Q}=99\%$. In Figure 10c,d DSLO-CNN converges faster and achieves a more stable performance with the increase of training scenarios. In all these cases, DSLO-DNN converges much slower than DSLO-CNN. Hence, the proposed DSLO-CNN is suitable for dynamic MEC scenarios with few training samples.

In Figure 11, we further statistically test the adaptability of DSLO to new MEC scenarios. We plot both the median and the confidence intervals of $\hat{Q}$ overall test scenarios. As shown in Figure 11a, we make the number of training scenarios 70% of the full scenario, as $\left|I_{train}\right|=70\%\left|I\right|$. We adjust the training set by changing the size of $\left|K_{train}\right|$ while keeping other parameters unchanged. When $\left|K_{train}\right|=1$, it means that there is only one training sample per MEC scenario $\left\{\Psi_{i}|i\in I_{train}\right\}$. The median of DSLO-CNN is greater than 98%, and the median of DSLO-DNN is no more than 96%. As the number of training samples increases, DSLO is more adaptable to new scenarios. In general, DSLO-CNN outperforms DSLO-DNN, especially with few training samples. As shown in Figure 11b, we evaluate DSLO that trained under different sizes of $\left|I_{train}\right|$. At the same time, the number of training samples is kept constant for each scenario, as $\left|K_{train}\right|=2$. When $\left|I_{train}\right|=10\%\left|I\right|$, the median of the DSLO-CNN algorithm is greater than 96%. When $\left|I_{train}\right|=70\%\left|I\right|$, we see that the median of the DSLO-CNN algorithm can reach more than 99% and the confidence intervals are mostly above 98%. In comparison, the median of the DSLO-DNN algorithm is always less than 98%. Under different scales of training sets, the adaptability of DSLO-CNN to new MEC scenarios is always better than that of the DSLO-DNN algorithm.

### 5.4. Comparisons with MELO and ARM

To further evaluate the effectiveness and superiority of the proposed algorithm, we compare the proposed algorithm with other computational offloading algorithms. In addition to the LR [33] algorithm, two different benchmark algorithms, MELO [11] and ARM [34], have been evaluated. MELO is a meta-learning-based computing offloading algorithm that learns a priori knowledge of historical MEC scenarios and quickly converges when encountering new MEC scenarios by fine-tuning training with only a few samples. ARM is an adaptive risk minimization framework, which extracts global features from different scenarios to update the network model and improves the model’s adaptability to unknown scenarios.

Before fine-tuning the training, the models were trained with a sufficient number of samples, where $\left|I_{train}\right|=70\%\left|I\right|$ as well as $\left|K_{train}\right|=4$. It is worth mentioning that the input of the MELO algorithm model does not contain weight information, and other parameters are set in the same way as the DSLO algorithm. To avoid chance, we use the average value of the fine-tuning test in 10 MEC scenarios as the actual test effect. As shown in Figure 12, the LR algorithm produces a constant normalized system utility value $\hat{Q}=0.934$. The MELO algorithm converges to the same performance as DSLO-DNN in 30 fine-tuning steps, resulting in a normalized system utility value $\hat{Q}$ over 98.5%. The ARM algorithm is implemented based on CNN and generates a normalized system utility value $\hat{Q}=0.99$, which is greater than MELO and DSLO-DNN. However, the normalized system utility value $\hat{Q}$ generated by DSLO-CNN is always greater than 99%. Hence, DSLO-CNN makes full use of the weight information and can achieve better model initialization parameters than other computational offloading algorithms with a small number of training samples. The algorithm can be quickly adapted to new MEC scenarios.

## 6. Conclusions

In this paper, we propose a deep supervised learning-based offloading algorithm, DSLO, which requires few training samples and can adapt to different MEC computing task scenarios. We take advantage of convolutional neural networks to efficiently handle multidimensional state parameters. The proposed algorithm can quickly adapt to dynamically changing MEC scenarios and achieve near-optimal offloading decisions. Numerical results have validated the accuracy of the proposed algorithm and the performance advantage compared with the existing MELO algorithm. Extensive numerical results are evaluated to study the performance of both DSLO-CNN and DSLO-DNN algorithms. DSLO outperforms the current benchmark algorithms in the metrics of system utility and CPU execution latency. In general, DSLO-CNN requires fewer training samples than DSLO-DNN. It can achieve 99% normalized system utility $\hat{Q}$ by using only four training samples per MEC scenario.

The proposed DSLO algorithm in this paper relies on a small size of labeled data, which limits its application in some MEC scenarios whose optimal labels are unavailable or difficult to be obtained. Moreover, DSLO requires all training data to be collected and trained in a centralized MEC server, which may raise some security and privacy issues related to personal data. For future work, we expect a DSLO algorithm implemented via reinforcement learning methods without manually labeled data. Furthermore, considering the data privacy in distributed MEC scenarios, implementing DSLO on a federated learning framework is interesting and necessary. Considering the heterogeneity and complexity of MEC networks, we expect that DSLO can be further extended to dynamically changing MEC scenarios with online learning. It will benefit future offloading algorithm deployment on large-scale MEC networks. 

## Figures and Tables

**Figure 1 sensors-22-04088-f001:**
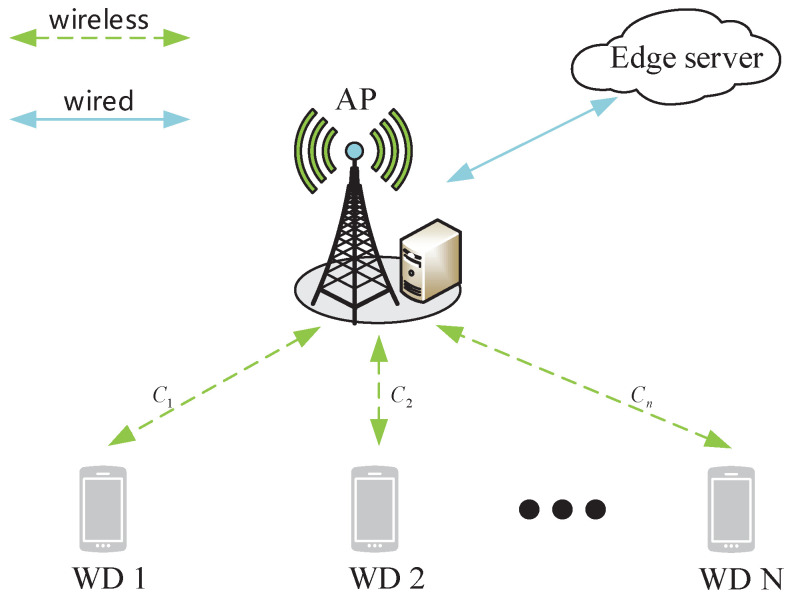
System model of an MEC network with multiple WDs.

**Figure 2 sensors-22-04088-f002:**
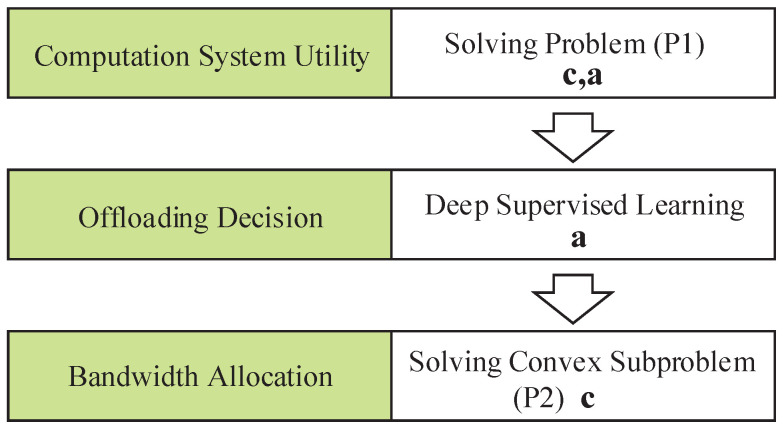
The two-level optimization structure for solving the problem (P1).

**Figure 3 sensors-22-04088-f003:**
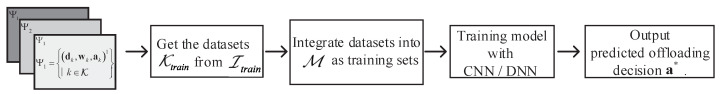
The pipeline of DSLO training.

**Figure 4 sensors-22-04088-f004:**
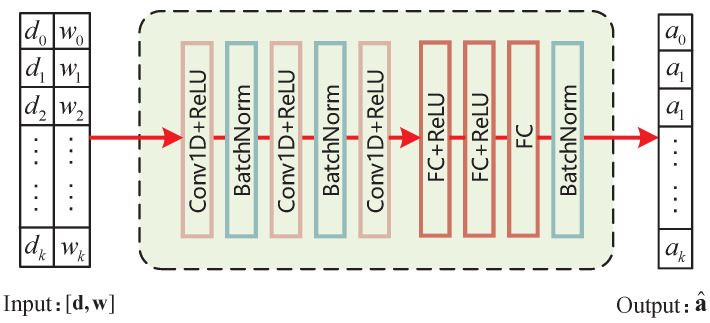
The process of the DSLO-CNN algorithm.

**Figure 5 sensors-22-04088-f005:**
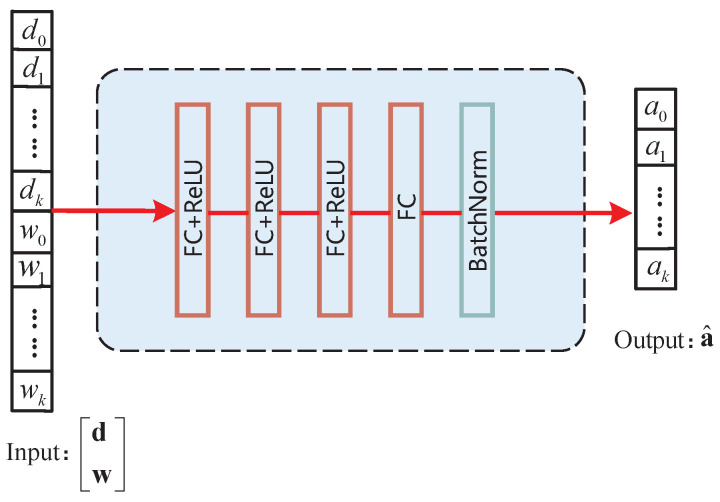
The process of the DSLO-DNN algorithm.

**Figure 6 sensors-22-04088-f006:**
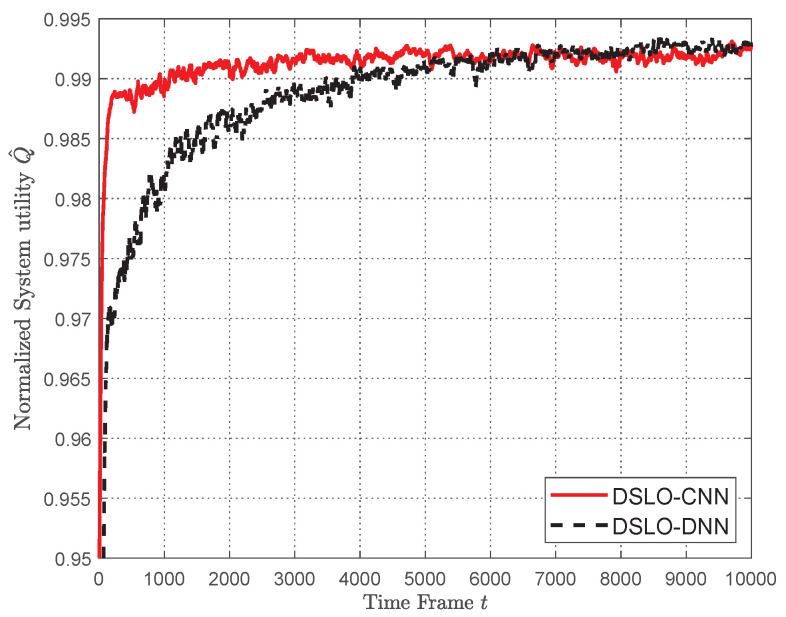
Convergence performance of DSLO with plenty of training samples.

**Figure 7 sensors-22-04088-f007:**
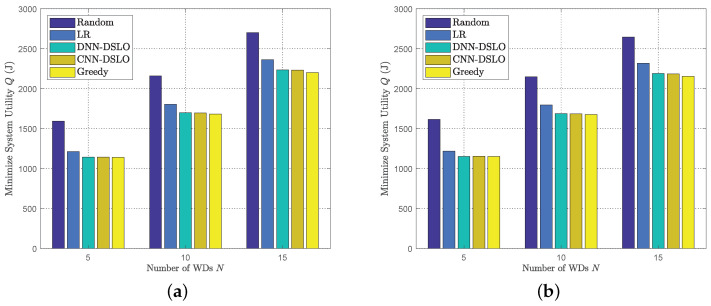
Comparisons of system utility performance for different offloading algorithms. (**a**) $\Psi_{1}$; (**b**) $\Psi_{2}$.

**Figure 8 sensors-22-04088-f008:**
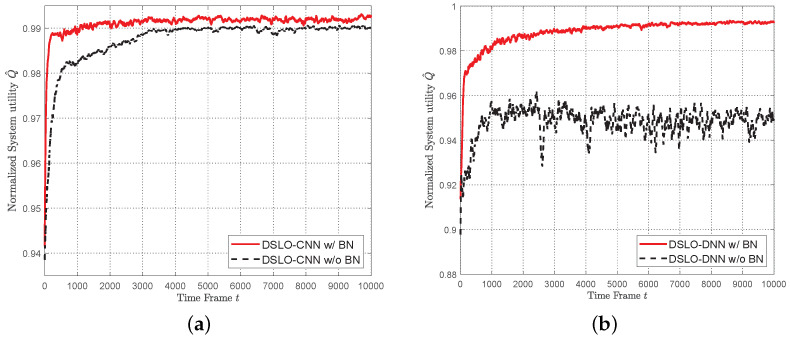
Performance evaluation of BN layer. (**a**) DSLO-CNN; (**b**) DSLO-DNN.

**Figure 9 sensors-22-04088-f009:**
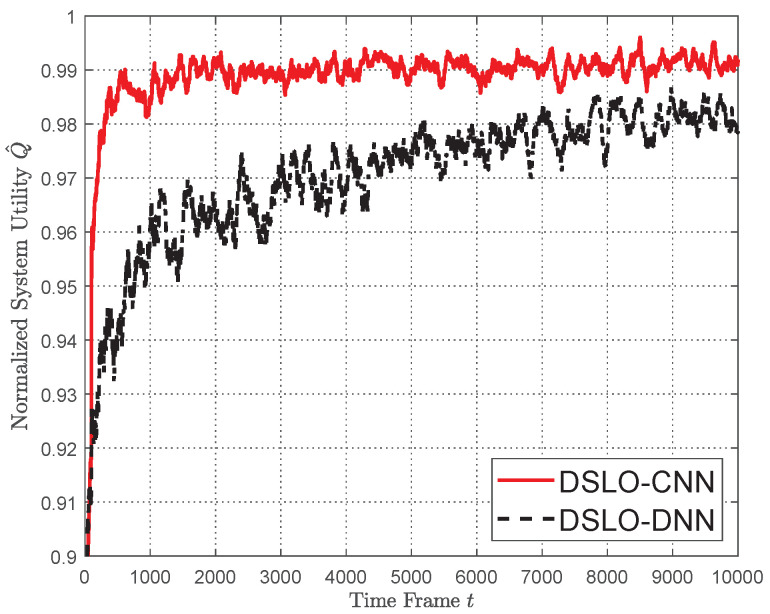
DSLO with $\left|K_{train}\right|=2$ per MEC scenario.

**Figure 10 sensors-22-04088-f010:**
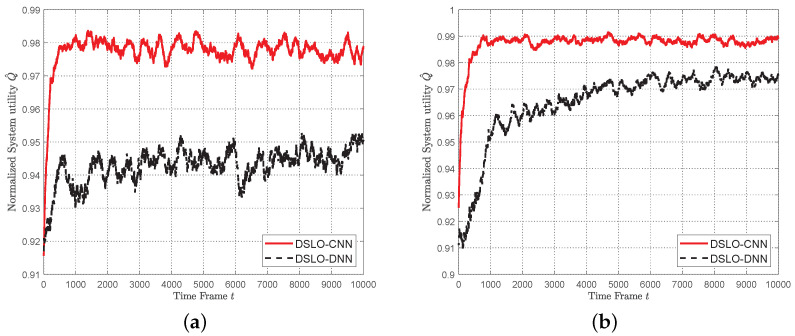
Convergence performance of DSLO under different scales of training MEC scenarios $I_{train}$. (**a**) 10%I]; (**b**) 30%I; (**c**) 50%I; (**d**) 70%I.

**Figure 11 sensors-22-04088-f011:**
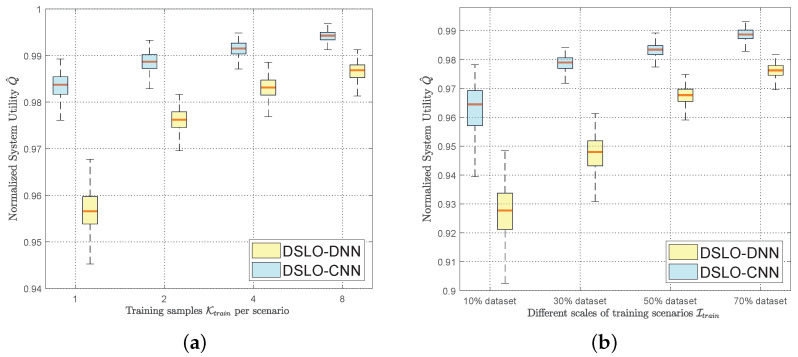
Test performance with different scales of the training dataset. (**a**) $\left|I_{train}\right|=70\%\left|I\right|$; (**b**) $\left|K_{train}\right|=2$.

**Figure 12 sensors-22-04088-f012:**
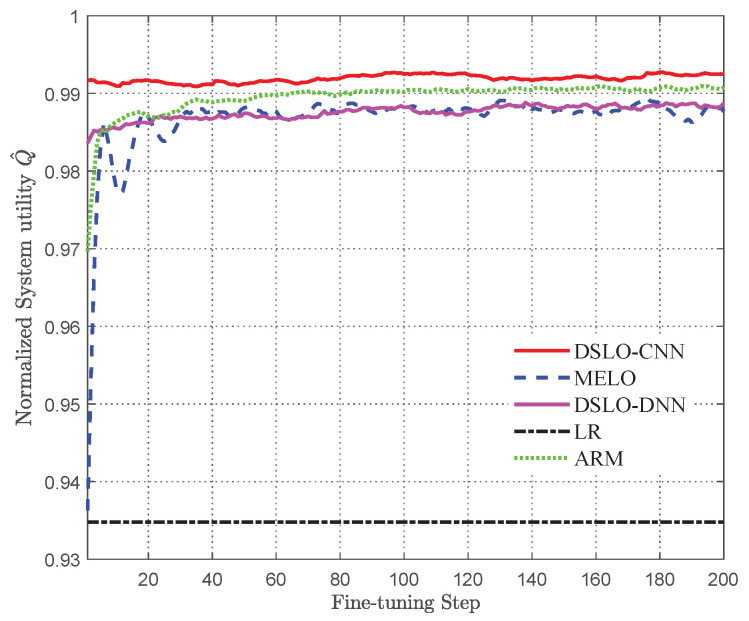
Test performance of different computational offloading algorithms.

**Table 1 sensors-22-04088-t001:** The parameters of DLSO-CNN and DSLO-DNN algorithm structures.

(a) DSLO-CNN Algorithm			
**Layer**	**Size**	**Activation**	**BN**
Conv1d1	16	ReLU	16
Conv1d2	16	ReLU	16
Conv1d3	3	ReLU	-
fc1	21	ReLU	-
fc2	64	ReLU	-
fc3	10	Sigmoid	10
**(b) DSLO-DNN algorithm**			
**Layer**	**Size**	**Activation**	**BN**
fc1	20	ReLU	-
fc2	120	ReLU	-
fc3	80	ReLU	-
fc4	10	Sigmoid	10

**Table 2 sensors-22-04088-t002:** Simulation parameters.

Notation	Value	Notation	Value
*C*	100 Mbps	$d_{n}$	10–30 MB
$e_{n}^{l}$	$3.25\times 10^{-7}$ s/bit	$t_{n}^{l}$	$4.75\times 10^{-7}$ s/bit
α	$3.5\times 10^{-7}$ J/bit	$\gamma_{n}$	1900 cycles/byte
$e_{n}^{t}$	$1.42\times 10^{-7}$ J/bit	CPU rate	$10\times 10^{9}$ cycles/s

**Table 3 sensors-22-04088-t003:** Weight factors of different WDs.

MEC Task Scenarios	Weight		
	*N* = 5	*N* = 10	*N* = 15
$\Psi_{1}$	{1.0, 1.5, 1.0, 1.5, 1.0}	{1.0, 1.0, 1.5, 1.5, 1.0 1.5, 1.5, 1.0, 1.0, 1.5}	{1.0, 1.0, 1.5, 1.5, 1.5 1.0, 1.5, 1.0, 1.5, 1.0 1.5, 1.5, 1.5, 1.0, 1.0}
$\Psi_{2}$	{1.0, 1.5, 1.5, 1.5, 1.0}	{1.0, 1.5, 1.0, 1.5, 1.0 1.5, 1.0, 1.5, 1.0, 1.5}	{1.0, 1.5, 1.0, 1.5, 1.0 1.5, 1.5, 1.5, 1.0, 1.0 1.0, 1.0, 1.5, 1.5, 1.0}

**Table 4 sensors-22-04088-t004:** Comparisons of CPU execution latency.

# of WDs	DSLO-CNN		DSLO-DNN		LR
	Train	Test	Train	Test	
5	$7.5\times 10^{-3}$ s	$1.4\times 10^{-4}$ s	$3.5\times 10^{-3}$ s	$1.4\times 10^{-4}$ s	$4.1\times 10^{-2}$ s
10	$8.3\times 10^{-3}$ s	$2.5\times 10^{-4}$ s	$3.8\times 10^{-3}$ s	$2.3\times 10^{-4}$ s	$1.4\times 10^{-1}$ s
15	$9.2\times 10^{-3}$ s	$3.4\times 10^{-4}$ s	$3.9\times 10^{-3}$ s	$3.4\times 10^{-4}$ s	$3.1\times 10^{-1}$ s

## Data Availability

Not applicable.

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
