# Peer review of "Deep Learning-Based Dynamic Computation Task Offloading for Mobile Edge Computing Networks"

_sensors, 2022, doi:10.3390/s22114088_

Round 1
Reviewer 1 Report
Deep Learning based Dynamic Computation Task Offloading for Mobile Edge Computing Networks
The authors propose a deep supervised learning-based computational offloading (DSLO) algorithm for dynamic computational tasks in multi-access / mobile edge computing (MEC) networks.
- The introduction of the manuscript has to be improved. There is a lack of completeness and continuity in the arguments of the introduction.
- The contributions of the manuscript should be better founded.
- A clear comparison with similar studies is missing.
- The authors have already published related articles. Therefore, a clear confront/contrast with those articles should be included.
- Justifications are missing in the description/settings of the simulations.
- Some inconsistency is present in the simulation results.
In addition, I have attached the manuscript with some parts highlighted in yellow that need to be revised.

Reviewer 2 Report
The article is an interesting example of using ML to solve problems in the area of network traffic management and load balancing. In principle the article is well written but there are some shortcomings resulting, in my opinion, from too poor description or lack of explanation of some problems in more detail. Below are selected comments that, in my opinion, will improve the quality of the article:
- throughout the paper, the authors introduce a number of variables that are not described in detail, e.g.: what does ε mean in section 2.1? enl - what does n mean what does l mean. I suggest that you go through the whole paper and check it carefully for missing descriptions.
- There are also a number of general assumptions at work. For example: "Similar to many studies, the size of the calculation result is usually much smaller than the size of the input" This is a rather risky assumption, but if we are to make it, it should be supported by appropriate citations.
- Some of the assumptions of the model require a more detailed description, or appropriate support by citations from the literature. For example, formula (4) calculates transmission delay in seconds dn is workload (in what units? % - dimensionless? ), cn to allocated bandwidth (in what units?). How dividing such quantities results in the quantity measured in seconds (ms) - this requires a broader explanation. Similar observations apply to other assumptions introduced in section 2.1. A good description of the model will allow to understand the whole article and the authors' own contribution.
- Section 3.2 provides a more extensive description of the proposed solution in which the authors use random sampling to train the model. Do the authors specify sampling rate limits and their effect on the results obtained?
- It would be useful to describe more extensively the form of record used for training and provide a sample tuple of pre-generated a dataset composed of various MEC task scenarios
In conclusion, the authors should carefully review the article for missing descriptions and description of the presented assumptions, which will improve its readability and highlight the authors' own contribution. At the moment the authors use many simplified descriptions without giving any sources in literature- maybe it is because this is another article in this thematic series. I believe that improving the article in this area will allow to fully assess the potential of the solution presented by the authors and will allow to obtain a recommendation for publication in a journal.
Reviewer 3 Report
- In the introduction section, the authors should highlight the challenges with the existing state of the art methods that motivated the need to introduce this work.
- Clearly define all the symbols presented in this work.
- In section 4, the authors have included the hyper-parameter settings. Would it be possible for the authors to dicuss about how did the authors obtain these values. Are these optimised values.
- Include an ablation study for understanding the performance of the proposed deep learning model.
- Compare the results of the proposed deep learning model with other possible baseline models and exisiting state of the art works.
- It is recommended to include the vector graphics images for all the figures.
- In the conclusions, briefly dicuss the limitations of the proposed model and also elaborate on the future prospects and opportunities of this research.
Reviewer 4 Report
Thank you for providing me with the opportunity to read “Deep Learning based Dynamic Computation Task Offloading for Mobile Edge Computing Networks”. I have the following comments:
- Please use the mdpi format where all lines are numbered. This helps reviewers provide line-specific comments.
- The paper needs serious language editing as it has serious grammar and syntax errors. For example, The first sentence of the abstract is very confusing and must be revised.
- The first line of the abstract has the word “computing” repeated twice. Please correct it.
- Before you discuss what the paper aims to do, please present the problem in the abstract.
- Add the implications of the work in a line towards the end of the abstract.
- Please reduce the number of references in grouped clusters [1-4], [8-13] etc., up to two references should suffice the requirements of justifications.
- Introduction, 2nd paragraph, need references to justify the claims. Please add 1-2 references here.
- There is a contradiction between the elements of abstract and introduction. While the abstract does not mention IoT at all, the introduction is very focused on IoTs. Please revise and make sure these are coherent and complement each other.
- Please separate the introduction and literature sections to focus on the latest state-of-the-art literature. Alternatively, you can rename section 1 as “introduction and background” and add more relevant literature there. Instead of reference clustering, please discuss the key takeaways and add a critical review of the literature to strengthen the quality of arguments in the paper.
- Move figure 1 to the method section where it is first mentioned.
- Please add relevant references to the method section for various equations and formulas adopted from other studies.
- Please move Figures 2, 3, 4, and others where they are first mentioned; otherwise, they appear to be haphazard and confusing.
- There are too many abbreviations in the paper. Please add a table of abbreviations to the paper and spell out all these in there.
- Since Table 1 has two separate sections, it is important to discuss them separately in detail. I suggest the authors combine them in a single table.
- Please properly discuss all components of each figure. For example, the components a, c, and d of figure 7 haven’t been discussed properly. These need clear and detailed discussions to highlight the key message. The same applied to components a and b of figure 8, and components a to d of figure 10.
- Figure 11 appears in conclusion, which doesn’t look appealing. Please move it before this section, where it is first mentioned.
- Please improve the discussions around the figures and tables. The authors should focus on the key takeaways to enhance the readability of the paper.
- The paper needs a detailed discussion section where the key takeaways of the paper are discussed. Further, it must be compared with other similar studies to highlight its novelties and key innovations. In the absence of this important section, the paper remains very weak. Please revise.
- Please clearly add the limitations of the study to the conclusion section.
- The future directions for expanding upon this study are not clear. Please revise
- Also, in 3-4 lines, discuss the study's practical implications and explain how it can help the practitioners.
Round 2
Reviewer 1 Report
Thank you for addressing most of my comments.
Author Response
Thanks for your comments and recognition of our effort in improving the quality of the manuscript.
Reviewer 2 Report
The authors of the article have taken my comments into consideration and have made applicable corrections to the article. I believe that in its present form the article can be published.
Author Response

(The authors gave the same response as above.)

Reviewer 3 Report
The authors have successfully addressed majority of the comments. Carefully proofread the manuscript.
Author Response
Thanks for your comments and recognition of our effort in improving the quality of the manuscript. We have proofread the revised manuscript.
Reviewer 4 Report
Thank you for addressing my comments. I have the following minor suggestions:
- Page 12 line 15, figure? Which figure? Figure 7? Always use the exact number for all figures when referring to them
- Ensure consistency across the paper for capitalization, the authors have used figure and Figure at various places. I suggest sticking to Figure followed by the number i.e. Figure 1, Figure 2, etc.
- Please compare the paper with add 1-2 more related papers in discussion section. Currently it is compared with only one paper (ref 11), this is not enough for high quality scientific articles.
- The authors have added future directions but limitations of the study are not clear. These must be added to the conclusion section.
